# Influence of Cortisol on the Fibril Formation Kinetics of Aβ42 Peptide: A Multi-Technical Approach

**DOI:** 10.3390/ijms23116007

**Published:** 2022-05-26

**Authors:** Alessandro Nucara, Francesca Ripanti, Simona Sennato, Giacomo Nisini, Emiliano De Santis, Mahta Sefat, Marina Carbonaro, Dalila Mango, Velia Minicozzi, Marilena Carbone

**Affiliations:** 1Department of Physics, Sapienza University of Rome, P. le A. Moro 5, 00185 Rome, Italy; nisini.1630469@studenti.uniroma1.it; 2Department of Physics and Geology, University of Perugia, Via Alessandro Pascoli, 06123 Perugia, Italy; 3CNR-ISC Sede Sapienza, Department of Physics, Sapienza University, P.le A. Moro 5, 00185 Rome, Italy; simona.sennato@roma1.infn.it; 4Department of Physics and Astronomy and Department of Chemistry-BMC, Uppsala University, Husargatan 3, 752 37 Uppsala, Sweden; emiliano.desantis@physics.uu.se; 5School of Pharmacy, Tor Vergata University of Rome, Via della Ricerca Scientifica 1, 00133 Rome, Italy; mahta.se2015@gmail.com (M.S.); dalila.mango@gmail.com (D.M.); 6Council for Agricultural Research and Economics (CREA), Research Centre for Food and Nutrition, Via Ardeatina 546, 00178 Rome, Italy; marina.carbonaro@crea.gov.it; 7Laboratory Pharmacology of Synaptic Plasticity, European Brain Research Institute, 00161 Rome, Italy; 8Department of Physics and INFN, Tor Vergata University of Rome, Via della Ricerca Scientifica 1, 00133 Rome, Italy; velia.minicozzi@roma2.infn.it; 9Department of Chemical Science and Technologies, University of Rome Tor Vergata, Via della Ricerca Scientifica 1, 00133 Rome, Italy; carbone@uniroma2.it

**Keywords:** Aβ42 peptide, fibril formation, ThT fluorescence, secondary structure, infrared spectroscopy, atomic force microscopy, molecular dynamics

## Abstract

Amyloid-β peptide (Aβ) aggregates are known to be correlated with pathological neurodegenerative diseases. The fibril formation process of such peptides in solution is influenced by several factors, such as the ionic strength of the buffer, concentration, pH, and presence of other molecules, just to mention a few. In this paper, we report a detailed analysis of in vitro Aβ42 fibril formation in the presence of cortisol at different relative concentrations. The thioflavin T fluorescence assay allowed us to monitor the fibril formation kinetics, while a morphological characterization of the aggregates was obtained by atomic force microscopy. Moreover, infrared absorption spectroscopy was exploited to investigate the secondary structure changes along the fibril formation path. Molecular dynamics calculations allowed us to understand the intermolecular interactions with cortisol. The combined results demonstrated the influence of cortisol on the fibril formation process: indeed, at cortisol-Aβ42 concentration ratio (ρ) close to 0.1 a faster organization of Aβ42 fragments into fibrils is promoted, while for ρ = 1 the formation of fibrils is completely inhibited.

## 1. Introduction

Amyloid peptides are addressed as primary causes of widespread neurodegenerative disorders, including Alzheimer’s, Parkinson’s, and type II diabetes diseases. The assembly of Aβ peptides into insoluble aggregates in healthy cells and membranes is considered as the most plausible cause of the Alzheimer’s disease (AD) onset in human brain [1]. One of the most studied and fascinating problems in biophysical and biochemical communities is the assembly of insoluble Aβ aggregates into fibrils, i.e., β-sheets stacked structures with long-range organization, whose size ranges from few to hundreds of nanometers [2]. Fibril formation has been studied in vitro with different phenomenological approaches and upon thermodynamical and compositional changes [3], but a complete understanding of the molecular processes is still lacking, as well as a clinical justification for Aβ polypeptide accumulation in cells and tissues. Further, several spectroscopic techniques are routinely employed to follow the Aβ aggregation in vitro; almost all of them provide a kinetics curve in line with the Boltzmann model for nucleation-dependent processes, where, after a variously lasting lag phase, an exponential growth occurs until saturation is reached [4]. A schematic representation is shown in Figure 1.

Despite the simplicity of the above model, the molecular mechanisms driving aggregation are difficult to disclose and are strongly dependent on concentration, pH, temperature, and incubation conditions. It is assumed that the lag phase is a preparatory stage for self-assembling nuclei with an antiparallel β-structure. Therein, the driving force for monomer packing is mainly hydrophobic and acts at the level of the monomer C-terminus [5]. Hydrophobic interactions are also addressed as the primary driving force during the elongation phase, where mature seeds pick up single Aβ peptides to form “polymers” (primary nucleation) after a modification of their secondary structure. Further, secondary nucleation, i.e., the formation of fiber replicas through oligomer-fiber interaction, is believed to speed up the growing processes in this phase [6]. However, not all oligomers evolve into fibrils, as neurotoxic aggregates can also be produced and remain stable in both the lag and the elongation phases [7]. Several efforts have been devoted to discovering amyloid aggregate inhibitors in the early stage of their formation, by exploring the use of different compounds (polyphenols, peptides, and nanoparticles) [8,9,10,11] and biochemical strategies [12], some of which are already in an advanced stage of clinical trials [13]. Currently, many synthetic and/or natural molecules are being studied as fibril inhibitors, but their biocompatibility is still far from being demonstrated. 

In this framework, it is also relevant to unveil the physical-chemical environmental conditions that could promote Aβ fibril formation in cells and membranes, whenever organic molecules (stressors), both of cellular and extracellular origin, act as catalytic agents. The fight-or-flight response of people working in extreme emergency conditions enhances the levels of their biochemical stressors, depending on the severity of the situation. Peaks of hormones, such as glucocorticosteroids, cathecolamines, prolactin, and growth hormones, are detected in the blood stream upon stress condition [14]. They are distributed and deposited in different organs of the body with collateral effects at different time scales. The immediate effects and the sudden cascade of reactions are the responses to counteract the stressful situations, whereas the longer-term aftermaths, such as the interaction with amyloids, tend to pile up through time with disease rising in elderly age [15,16,17,18,19,20]. In this scenario, also short- to medium-term effects are to be foreseen, as peaks of stressor concentration can accelerate degenerative effects. In addition, epidemiological studies indicate the enhancement of stressor adverse effects in conditions like depression [21,22]. At the molecular level, under stress conditions, cortisol, a glucocorticosteroid, is typically released; even if regulation mechanisms prevent cortisol accumulation, such as those based on norephinephrine [23,24], excess of this hormone might seldom be revealed in human blood, with consequences on proteins and a likely step up of Aβ peptide fibril formation. Moreover, cortisol excess, either as peak concentration in stress episode or as high basal level, is often correlated to early onset of AD [25,26]. Cortisol level in blood sensibly varies depending on the circadian rhythm and stress condition: the average levels at rest are reported in the 0.11–0.39 μM range, the lowest value achieved at 4:00 am. Such values increase to 0.25–0.85 μM at daytime, between 8:00 and 10:00, to gradually decline throughout the day [27]. In stress periods, cortisol concentration increases to 1.44 ± 0.86 μM in female students and 1.39 ± 0.86 μM in male students [28]. Spike cortisol concentrations, related to sudden stressful events such as fear episodes, should be also considered. So far, the mechanism of interaction between this hormone and proteins is not clear and has not been much studied yet. Therefore, it is essential to understand its functional role in the formation of protein aggregates.

In the present work, we investigated the role of cortisol in the in vitro kinetics of the Aβ42 fibril formation process at different cortisol-Aβ42 concentration ratios (ρ in the following), exploring a cortisol concentration range extending from high basal to peak levels. We remark that in vitro spectroscopy experiments require solute concentrations in a well-defined range, which, in most cases, are outside the range of the physiological concentrations. However, it is widely accepted to perform spectroscopic studies on fibrillation at “spectroscopic” rather than “physiological” concentrations [29]. Indeed, Aβ42 concentrations in plasma and cerebrospinal fluid are between 10^−6^ and 10^−4^ μM [30], thus well below the detection limit of the spectroscopic experiments and with a physiological ρ always greater than 1. Here, we explored a range of ρ values that extended from 0.1 to 1: the highest values mimic the physiological ones and provide the maximum degree of hormone-Aβ42 interaction as a limiting case of very dense solutions. On the other hand, the lowest values are more significant for the inner mechanisms of interaction between individual molecules. We thus studied the Aβ42 aggregate formation processes at different relative cortisol-Aβ42 concentrations with a multi-technical approach. Classical Molecular Dynamics (MD) simulation provided results of the early stage of the peptide-cortisol interaction, which allowed us to better unravel the intermolecular interactions at the atomistic level. InfraRed (IR) spectroscopy was also exploited to characterize the secondary structures of native and aggregate peptides: indeed, antiparallel and parallel β-structures are predicted for oligomers and fibrils, respectively [31]. However, the β-sheet IR markers cannot clearly discriminate between inhomogeneous aggregates (oligomers) and mature fibrils, apart from spectral differences that are tricky to observe [32]. Consequently, the IR spectra were here used to characterize specific stages of the Aβ42 path to fibrils, not to reconstruct the whole aggregation path. Likewise, Atomic Force Microscopy (AFM) images provided an insight on the morphological properties of mature fibrils and aggregates at some specific steps along the fibril formation path. The time course of aggregate formation at different ρ values was followed by a Thioflavin T (ThT) fluorescence assay.

Our results point towards a drastic shortening of Aβ42 fibril formation kinetics in the low ρ range, conferring a greater availability for one kind of fibril growth to the formed cortisol-Aβ42 complexes. The observed faster process also corresponds to a speed up of the elongation phase in which mature Aβ42 fibrils are formed, as a signature of a more effective grow-up mechanism. Such phenomenology frames into an enhanced activity of the secondary nucleation mechanism, powered by a greater number of initial seeds formed by cortisol and prone to replicate. Instead, the outcomes in the opposite limit of high ρ values provide evidence of inhomogeneous aggregates, being the formation of Aβ42 fibrils completely hindered by the cortisol excess. 

## 2. Results

### 2.1. IR Characterization of Aβ42 and Cortisol-Aβ42 Solutions

Infrared analysis was performed for Aβ42 in 4PBS:1DMSO buffer, ensuring that absorptions from cortisol and PBS do not alter that of the Aβ42 peptide in the amide I and II spectral region. The amide I absorption bands of Aβ42 at different stages of incubation are reported in Figure 2. Figure 2A refers to sample in its initial state before incubation: here, the most intense band centered around 1650 cm^−1^ reflects the α-helix character of the peptides, together with minor contributions from the turns (1670–1680 cm^−1^). Moreover, signatures of β-secondary structures were also detected as the contributions centered at 1625–1630 cm^−1^ and 1695 cm^−1^, in agreement with previous IR data [33]. This finding indicates that Aβ42 peptides quickly form oligomers, which are energetically favored in β-structured configurations. The presence of oligomers in the native state is not at all surprising, since it has been demonstrated that the encompassed 11–24 and 28–42 residues have strong affinity to form pairs of strands in hairpin configuration [34]. A global fit to second derivative data and spectra with Gaussian lines provided an estimate of the secondary structure percentages in the native peptides. Considering the effectiveness of the transition dipole strength in the different secondary structures [35], we found that most of the intensity (80%) is assigned to the peak at 1648–1655 cm^−1^, which originates from the superposition of the α-helix and the random coil secondary structures. An amount of 6% is obtained for β-secondary structures and one of 14% is provided for turn structures. These percentages are compared with those measured in Aβ42 peptides dissolved in DMSO (data not shown), which returned 88% of α-helix and/or random coil secondary structures, 8% of β-secondary structures, and 4% of turn structures. Details of fits and related secondary structures are reported in Appendix A and Appendix A of Appendix A.

The amide I absorption of Aβ42 incubated in the same buffer for 3 h is reported in Figure 2B. The overall shape of the spectrum consists of two main components: one centered at 1644 cm^−1^ accounting for random coils structures, and a second band detected at 1695–1705 cm^−1^, which is the marker of β-antiparallel structures, both from fibrils and oligomers. In this case, it is worth noticing that the spectrum of incubated peptide in DMSO buffer shows a high frequency band but red-shifted by few wavenumbers. Further, less intense and unresolved bands are detected at 1625 cm^−1^ and 1690 cm^−1^, ascribed to β-strands and to turn and/or β-antiparallel contributions, respectively. The Gaussian global fit (Appendix A) returns 75% of unordered secondary structure, 16% of β-strands are either parallel or antiparallel, and 9% of the intensity is due to the contribution centered at 1690 cm^−1^. We are led to interpret this spectrum as an intermediate or incomplete stage of fibril formation, where most of the aggregates did not achieve stable β-arrangements and oligomers still were in an unfolded or disordered state. On the contrary, the amide I spectrum shown in Figure 2C and referring to the Aβ42 sample incubated for 20 h is well-resolved in two intense β-secondary structure components in the 1625–1633 cm^−1^ and 1695–1700 cm^−1^ regions, respectively. The fitting procedure returns 38% of the intensity for the former band and 62% for the high frequency contribution. This “all β-structure” spectrum labels the final stage of protein aggregation kinetics. Once again, analogous, red-shifted bands were observed for Aβ42 peptides after incubation in DMSO buffer. The low-frequency band originates from the superposition of the B(π,0) (parallel sheets) and B_2_ (antiparallel ones) vibrational modes, while that at highest wavenumbers is ascribed to the B_1_ vibrational mode of antiparallel sheets [36]. In view of the intensity ratio for these bands reported in [31], we conclude that most of Aβ42 peptides stabilizes into oligomers with antiparallel β-secondary structure. 

A similar analysis was performed for the cortisol-Aβ42 complex in 4PBS:1DMSO buffer. The spectrum of the complex before incubation is reported in Figure 2D. This spectrum differs from that in Figure 2A by an increase of β-contributions at the expense of the α-helix ones: the β-sheet bands at 1625 cm^−1^ and 1693 cm^−1^ account for 17% of the total intensity, while that around 1652 cm^−1^ (α-helix and turn structures) only accounts for 58%. The loss of Aβ42 native secondary structure is ascribed to the effect of cortisol and more oligomers than native Aβ42 peptides are formed. The amide I spectrum of the cortisol-Aβ42 complex after 8 h of incubation is reported in Figure 2E: the contribution from α-helix structures is hardly detectable (around 1% of the total intensity), while the overwhelming fraction of intensity (89%) is assigned to random coil structures, and the remaining 10% is up to the β-structures. Finally, the amide I band of the complex after 23 h of incubation is shown in Figure 2F: this spectrum looks quite similar to that in panel E, but a fit to data reveals a larger amount of random coil structures. It is noteworthy that the increase of the contribution at 1667 cm^−1^ is likely due to turn structures.

The IR spectra of the cortisol-Aβ42 complex establish the presence of many unordered peptides, even at the early stages of incubation, with identical secondary structures in short or long incubation times. The lack of the well-resolved band at 1695–1705 cm^−1^ detected in Aβ42 at the longest incubation times suggests that, in this case, the aggregates are more prone to form unordered structures and that fibrils, instead, grow in β-parallel stacked structures, which are likely present even at the shortest incubation times.

Spectra at the initial and final stage of incubation (20 h) of the cortisol-Aβ42 complex at ρ = 1 are reported in Figure 3. The spectral shape of amide I of both samples shows a major band assignable to unordered structures. The small high-frequency contribution (1700 cm^−1^) detected after 20 h of incubation denotes a marginal presence of β-antiparallel structures.

### 2.2. Molecular Dynamics Simulation of Cortisol–Aβ42 Complex

In order to address information on the intermolecular interactions, we performed MD simulations of two model systems, each composed by five Aβ42 peptide monomers in water either in the absence (ρ = 0) or in the presence of five cortisol molecules (ρ = 1). For each system, three replicas were simulated, as described in detail in the Materials and Methods section. The final configurations of the five Aβ42 peptides in the absence (upper row) and in the presence (lower row) of cortisol are shown in Figure 4. In the short time of the simulations, the analysis of the secondary structure of Aβ42 peptides provided a total percentage of around 65% for helix and random coil structures and of 5% for β-structures (Appendix A, other secondary structure components are not reported). The latter percentage smoothly increases over the simulation time for the systems in the absence of cortisol. This behavior meets the IR outcomes on the parallel and antiparallel β-arrangements of aggregates and oligomers.

MD results reported in the lower row of Figure 4 point out the larger affinity of Aβ42 peptides to aggregate when bonded with cortisol, as the formation of a single complex is observed in all the simulated replicas. The presence of cortisol also seems to speed up aggregate formation (Appendix A). Further, no drastic changes of the secondary structure of the peptides bonded to cortisol were foreseen by MD, apart from a slight decrease of the helix ones. However, it is remarkable that the β-structure percentage does not change with increasing time, suggesting that the bond with cortisol hampers the capability of the peptides to convert in β-arranged nuclei for seeding fibrils.

The MD results also allowed us to infer the preferential binding sites of cortisol molecules on the Aβ42 peptides: we considered whether cortisol molecules prefer to interact with amino acids whose side chains share specific chemical properties. Our results (reported in Appendix A) do not provide evidence of privileged side chains as binding sites, returning identical probability for cortisol to interact with each amino acid of the Aβ42 peptide. According to this finding, we assume that the binding between Aβ42 and cortisol occurs through the formation of hydrogen bonds (H-bonds), since the polar and the hydrophobic/hydrophilic nature of the residues seems ineffective. The cortisol molecule has, indeed, a hydrophobic nature, but it is also prone to form up to eight H-bonds (five as the acceptor and three as the donor). Therefore, we hypothesize that an early contact between peptides and cortisol occurs through long-range hydrophobic interactions, followed by the formation of short-range H-bonds, according to a multistage bonding kinetics that is reminiscent of those often observed in membrane-protein systems [37]. This hypothesis is confirmed by the analysis of the time evolution of the number of H-bonds among Aβ42 peptides and cortisol molecules reported in Figure 5: the number of H-bonds increases with simulation time, and it reaches a plateau value of about 22 in the last 50 ns of simulations. This number indicates that every cortisol molecule forms, on average, 4.4 H-bonds with every Aβ42 peptide.

### 2.3. ThT Fluorescence Assay

To understand the kinetics of Aβ42 fibril formation process both in the absence and in the presence of cortisol, we measured the ThT fluorescence emission in the 450-600 nm wavelength range. ThT fluorescence spectra during the incubation time of native Aβ42 peptide, of the cortisol-Aβ42 complex at ρ = 0.1, and at ρ = 1 are reported in Figure 6. In all samples with ρ < 1, the maximum fluorescence intensity appears in a shorter time with respect to the ρ = 0 case (Figure 6A). However, the fluorescence intensity measured in the saturation phase exponentially decays with the concentration ratio (inset of Figure 6C), proving the formation of a lower number of fibrils in these samples. Remarkably, after 85 h of incubation, the ThT signal undergoes a significant damping both for ρ = 0 and for ρ = 0.1 (see the dotted spectra in Figure 6A,B), whose origin does not have a clear explication yet. We observed (see next section on AFM measurements) that long mature fibrils are present when maximum fluorescence signal is detected in the saturation phase, whereas much shorter structures are observed in samples where the signal is damped. Even assuming that the effectiveness of ThT dye is higher for the longest structures, we did not find a straightforward cause for fibrils to split into smaller moieties at the longest incubation times.

The area subtended by the spectra is reported in Figure 7A once it is scaled to one. For each point, the error was estimated by repeating the experiments in nominal identical conditions. The residuals obtained from a Boltzmann fit to data provided the maximum uncertainties to be assigned on the points. An exhaustive example of reproducibility is shown in Appendix A.

Interestingly, the sample with the lowest cortisol amount (ρ = 0.1) is more responsive to incubation, and the fibril formation process is speeded up. On the other hand, at the highest concentration ratio (ρ = 1), the ThT fluorescence is negligible for the whole incubation time (gray curve), which indicates the hindering of the mature fibril growth. The sigmoid curves used to fit the experimental data provide two parameters: $t_{onset}$ (calculated at the starting point of the exponential phase) gives advice of the lag phase duration, and $t_{elongation}\;$($t_{elongation}=t_{half}-t_{onset}$, with $t_{half}$ the inflection point of the sigmoid) marks the rapidity of the avalanche processes. The former quantity is reported in Figure 7B as a function of the concentration ratio (apart for ρ = 1), with errors estimated as the maximum deviation from the average of repeated measurements. We observe a drastic shortening by a factor 6 of the lag phase for ρ = 0.1 with respect to ρ = 0; instead, for ρ = 0.2 and ρ = 0.3, the lag phase is reduced by a factor 3. We suppose that at higher ρ values (0.3 < ρ < 1, still not characterized), the lag phase duration recovers similar or even higher values to those of the native Aβ42 peptide, with a significantly reduced number of mature fibrils. Moreover, we note shorter elongation time values at the lowest concentration ratio; this outcome suggests that one specific exponential mechanism becomes more effective over others competing kinetics.

### 2.4. AFM Morphological Analysis

The most representative samples studied by ThT fluorescence assay were further analyzed with AFM to figure out possible morphological changes among samples at different ρ values. Three different images of Aβ42 after incubation are reported in Figure 8; they represent the three most frequently observed patterns in the native Aβ42 sample after different incubation times.

Figure 8A shows an image of fibrils formed after 40 h of incubation; an accurate inspection (see Appendix A for the statistical analysis) reveals a few μm long and about 5 nm high fibrils, which are formed by interlinked fibers. The image in Figure 8B represents one cluster of very short and thin fibrils detected in Aβ42 sample after 78 h of incubation; the statistical analysis on a collection of images confirms that the average height and length of fibrils are around 1 nm and 250 nm, respectively. AFM images thus confirm the presence of shorter and thicker fibrils at the longest incubation times, as suggested by the ThT assay. The dense pattern visible on the background of the same image can be ascribed both to oligomers and to thicker aggregates with average height of 2 nm, which are visible as brighter spots and worm-like structures. Moreover, in the same Aβ42 sample, different kinds of aggregates, as worm-like structures isolated or superimposed to the longer fibrils, were observed (Figure 8C). Their height is on average about twice the ones in panel B; in particular, in the image in panel C, three main structures of 2 nm, 3 nm, and 7 nm thickness are resolved by the statistical analysis of the height. AFM images collected at the end of the incubation for samples at ρ = 0.1 (Figure 8A,B, after 24 h of incubation) and ρ = 1 (Figure 8C, 60 h) are reported in Figure 9.

We focused on these samples as the most significant in the ThT kinetics assay. For ρ = 0.1, the AFM inspection reveals extremely thick single fibers, whose height is less than 1 nm (Appendix A and Appendix A), diluted in a pattern of unstructured oligomers. These findings confirm the indications of IR and ThT fluorescence experiments, which foresaw a great number of unstructured oligomers and a minority of fibrils. On the contrary, the sample with ρ = 1 does not present fibrils, either single strands or twinned fibers, but only huge isles of aggregates, an order of magnitude thicker than those at ρ = 0.1. The newness of the AFM results resides on the different morphology of the fibrils observed in the absence and in the presence of cortisol: in the latter case, their thickness does not exceed one nm, suggesting that the interaction with cortisol privileges the formation of shorter and thicker fibrillar structures even at the earlier incubation times.

## 3. Discussion and Conclusions

Our experiments provide a complex overview of kinetics and behaviors for fibril formation in the cortisol-Aβ42 complexes. The ThT assay disclosed the fibril formation pathway of native Aβ42 (ρ = 0), which follows a sigmoid curve with characteristic times ($t_{onset}=30\;h$ and $t_{elongation}=10\;h$) consistent with a model of superimposed kinetics [6]. At the same concentration ratio, the IR absorption spectra clearly showed Aβ42 secondary structure changes during incubation, from an α-rich phase in the native stage to a β-rich one; in addition, the presence of a meaningful number of unordered oligomers, potentially prone to seed fibrils, were foreseen at the intermediate incubation times. AFM inspection of the same sample exhibits the presence of both long and short fibrils, the former probably composed by the recruitment of monomers, the latter, on average less thick, suggestive for replication mechanisms.

The cortisol-Aβ42 complex kinetics at ρ < 1 is much faster, as $t_{onset}$ and $t_{elongation}$ values are much shorter than those observed at ρ = 0, but with a number of mature fibrils reduced by cortisol. The IR data of the cortisol-Aβ42 complex at ρ = 0.1 reveal that the parallel β-sheet and unordered secondary structures are the most likely conformations for the peptides in the complex after a relative short incubation time. AFM images acquired at the lowest cortisol relative concentration confirm the occurrence of very thick and untwined fibrils. In the high concentration limit (ρ = 1), all the experiments confirm the inhibition of the complex to form fibrils, but only aggregates with a high content of disordered secondary structures are observed. Besides, MD simulations show that the presence of cortisol seems to favor the formation of larger complexes in a shorter time scale. The MD results also give hints about the dynamics of cortisol-Aβ42 interaction, suggesting that a first hydrophobic interaction is followed by the occurrence of H-bonds, which might prevent the process of turning into β-structures.

To rationalize these outcomes, we consider the role of the cortisol-Aβ42 complex in solution. At ρ = 0, the simultaneous presence of both homogeneous primary nucleation and secondary replication can be inferred, as also suggested by literature, even if replication is predicted to account for only the 10% of the aggregation events [38]. On the contrary, the lack of fibrillation observed at ρ = 1 can be interpreted by assuming the formation of inhomogeneous aggregates from saturated or partially saturated complexes, within a framework of H-bonds, hydrophobic, and electrostatic interactions. This occurrence hampers the formation of seeds suitable for primary elongation and also inhibits the secondary nucleation processes. This hypothesis matches the AFM outcomes as, at the highest cortisol-Aβ42 ratio, only large heterogeneous aggregates are observed. In a straightforward model composed by five host molecules in a cubic box (12 nm side) with molecular sites of equal affinity for the guest molecule, a percentage around 5% of hosts are saturated (i.e., all binding sites are occupied), 35% are not saturated, and percentage of free hosts is around 60% at an equimolar concentration of guest molecules. Therefore, a small oligomer formed by ten Aβ42 peptides contains on average five cortisol molecules, sufficient to hamper the capabilities of the oligomer to seed β-strands. In the opposite limit, for ρ = 0.1, the same 10-peptide oligomer contains on average only one cortisol molecule.

The phenomenology observed at the lowest ρ values is less intuitive, and a complementary role of the cortisol-Aβ42 complex is expected to occur. The reduction of $t_{onset}$ observed at these ρ values can be ascribed to the presence of either a larger number of nuclei or to more active ones. The readiness of the fibrillar transition observed at the lowest ρ values suggests the necessary presence of a replication process. We postulate that the fibrillation nuclei at low cortisol-Aβ42 ratios are able to enhance seeding effects for both primary and secondary nucleation. A recent theoretical approach to Aβ42 fibril formation [38] provided a model and constraints for the enhancement of secondary nucleation: peptides less prone to bind with preformed fibrils, when involved into oligomers, can detach the oligomer itself from the fibril and establish a matrix for secondary replication. In this framework, we suggest that cortisol in a low relative concentration ratio might transform Aβ42 oligomers unsuitable for secondary nucleation to those prone to self-replication. According to the above cited theoretical models [38], a free energy difference of about 1 KT is necessary to transform an intermediate peptide suitable for secondary nucleation.

In conclusion, our experiments disclose a double role of the cortisol molecule when bonded to the Aβ42 peptide, based on the concentration ratio of the two molecules. We recall that, under physiological conditions, cortisol is present in plasma and in CSF at concentrations around 100 ng/mL and 6 ng/mL, respectively. The same evaluation for Aβ42 peptides provided values at least 10 times smaller (20 pg/mL in plasma, 554 pg/mL in CSF). In the high relative concentration regime, which mimics physiological conditions, the cortisol-Aβ42 complex forms highly inhomogeneous aggregates and hinders the formation of fibrils. This result is relevant as it would address biochemical and medical research towards the toxicity of aggregates formed in hormonal over-exposition condition. The second character of cortisol is no less harmful since it behaves as an effective stressor at the lowest concentration ratio. The present findings support previous studies on the correlation between elevated glucocorticoid levels found in AD patients, in particular those where over-expression of cortisol has been observed in mouse-model mouse models of AD [39] and at presence of early AD patients [40]. The mechanisms acting on the molecular scale are still far from being disclosed and further theoretical and experimental efforts are necessary to achieve a complete understanding. However, these preliminary data suggest a strong correlation between complex at low cortisol-Aβ42 ratio and the formation of seeds for secondary nucleation mechanisms.

## 4. Materials and Methods

### 4.1. Sample Preparation

Mixtures of cortisol and Aβ42 protein (purchased from Genscript Biotech, Piscataway, NJ, USA) were prepared starting from the initial solutions of 500 µM Aβ42 in DMSO and 500 µM cortisol in 4PBS:1DMSO. Different amounts of cortisol and protein solution were combined to get a volume of 12 µL, which was then added to 1188 µL of 4PBS:1DMSO in order to get a final volume of 1.2 mL for all samples. All the quantities used for the preparation are reported in Table 1. Samples were then incubated at 37 °C under stirring [41]. All the chemical reagents were purchased from Si Sigma Aldrich- Merck KGaA (USA).

### 4.2. Fluorescence Spectroscopy

Fluorescence measurements were performed with a Perkin Elmer fluorimeter in the 460–600 nm wavelength range using λ = 440 nm as excitation wavelength at different incubation times. For each measurement, 150 µL of a freshly prepared ThT solution in 4PBS:1DMSO was added to 150 µL of each sample (ThT: Aβ42 2:1 molar ratio). The fluorescence integrated intensity can be used as an estimation of the quantum yield of the fluorophore, which increases if the number and dimensions of fibrils increase. In this work, we considered the integrated area of the fluorescence spectra in the whole range. The resulting values were fitted through a sigmoid curve described by Boltzmann function: $y=\frac{A_{1}-A_{2}}{1+e^{\left(t-\tau \right)/dt}}+A_{2}$, where y represents the total integrated area, and τ is defined as the difference between $t_{onset}$ and $t_{elongation}$.

For a clearer comparison among data acquired at different Aβ42 final concentration, data and respective fitting curves were normalized in the 0–1 intensity range. Therefore, the different fibril formation onset times and kinetics are easier to evaluate.

### 4.3. Molecular Dynamics

We performed classical MD simulations of three replicas of two model systems each composed by five Aβ42 peptide monomers (pdb-id: 6szf) in water either alone or in the presence of five cortisol molecules (corresponding to ρ = 1 concentration ratio). In the starting configurations, both Aβ42 peptides and cortisol molecules were randomly placed in a cubic simulation box with a 12 nm side, filled with water and an appropriate number of counter-ions (necessary to get system neutrality). The simple point charge (SPC) model was used for water molecules. The system composition details are presented in Table 2.

Simulations were carried out by making use of the GROMACS package [42] employing the GROMOS96 54a7 force field [43]. The cortisol structure and force field were downloaded from the Automatic Topology Builder (ATB) repository [44,45]. Periodic Boundary Conditions (PBC) were used for calculations. Amino acid protonation states were modeled to resemble neutral pH. The simulation procedure (for each replica of the two model systems) was started by minimizing the potential energy of the whole system using the steepest descent algorithm. After that, each model system was equilibrated for 200 ps in the NVT ensemble at 150 K. Then, the production run, for each replica of each system, was a 200 ns long simulation in the NpT ensemble at 300 K. The temperature was held fixed at 300 K using the v-rescale thermostat [46] with coupling time of 0.1 ps. The pressure was kept at the reference value of 1 bar, exploiting the features of the Berendsen [46,47] with a coupling time of 1 ps and an isothermal compressibility of 4.5 × 10^−5^ bar^−1^. The Particle Mesh Ewald algorithm was employed to deal with the long-range Coulomb interactions [48]. A time step of 2 fs was used. A nonbond pair list with a cutoff of 1.0 nm was created and updated every 10 steps. The analysis of the collected MD trajectories was carried out by using both standard GROMACS tools, VMD tools [49] as well as some ad hoc written python scripts using the MDAnalysis package [50,51].

### 4.4. AFM Morphological Analysis

We recovered the samples at the end of the fluorescence experiments. A drop of 10 µL of each sample was deposited on freshly cleaved mica and gently rinsed by Milli Q water after 1 min of treatment. AFM measurements were performed by a Dimension Icon (Bruker AXS) instrument. AFM images were acquired in air at room temperature by employing Tapping Mode and RTESP-300 (Bruker) probes (10 nm nominal radius of curvature). AFM data processing was performed by Gwydion software. Background subtraction through line-by-line or polynomial data leveling of height was applied.

### 4.5. IR Spectroscopy

The absorption spectra were acquired with a Bruker IFS66VS Michelson interferometer, equipped with DTGS detector and working at low vacuum to reduce the atmospheric contributions to the spectra. The acquisition parameters (apodization, zero filling, and mirror path length) were chosen in order to achieve a spectral resolution of 2 cm^−1^. A high throughput transmission cell, sealed with CaF_2_ windows and designed to work under vacuum, was used to host the samples. The optical path length was estimated by interference fringes and adjusted at 1 µM. One drop of 4 µL was withdrawn from the Aβ42 (c = 500 µM) and DMSO solution for any assay. Experiments were repeated three times. OPUS software was chosen for the preliminary data analysis and the second derivative calculation, while codes for global fits with Gaussian lines were developed in house.

## Figures and Tables

**Figure 1 ijms-23-06007-f001:**
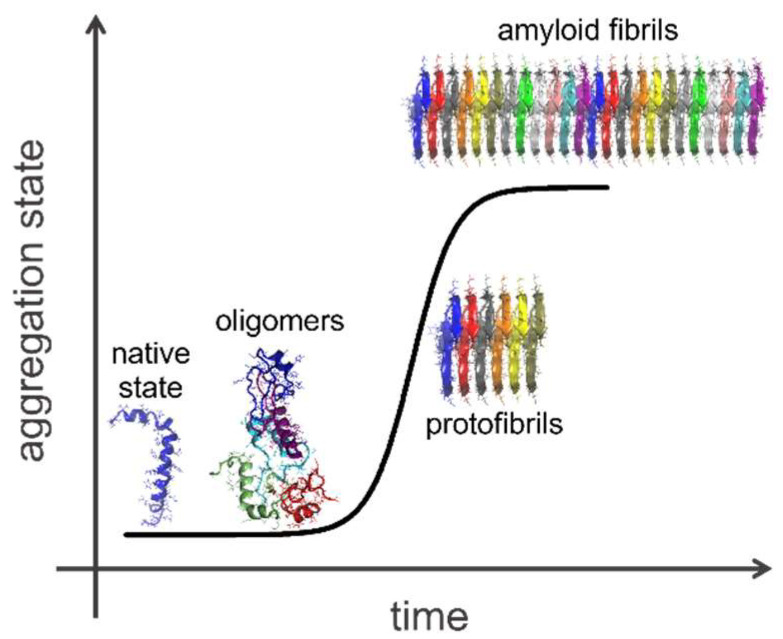
Scheme of temporal evolution of fibril formation upon nucleation processes.

**Figure 2 ijms-23-06007-f002:**
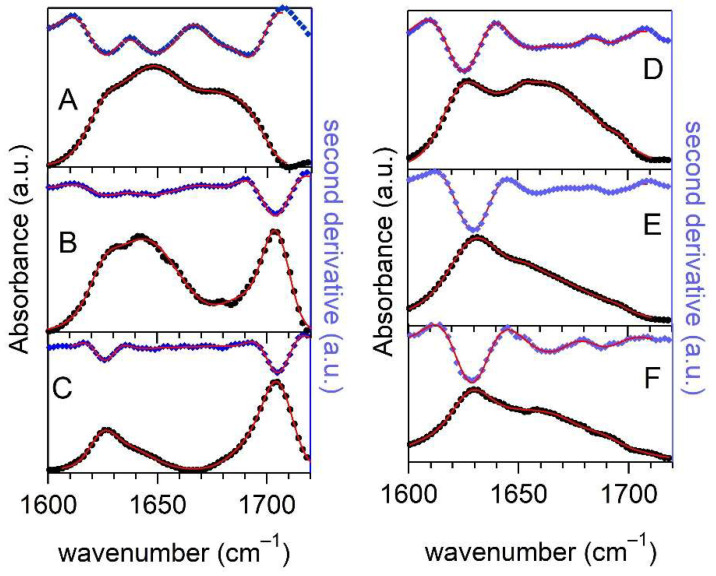
Amide I IR absorbance spectra (black) and their second derivatives (blue) of Aβ42 at c = 416 μM in 4PBS:1DMSO solution: (**A**) native state; (**B**) intermediate state after 3 h of incubation; (**C**) final state after 20 h of incubation; and final state of cortisol-Aβ42 complex (ρ = 0.1, Aβ42 concentration = 455 μM) in 4PBS:1DMSO buffer at different incubation times at 37 °C: (**D**) t = 0 h, (**E**) t = 8 h, and (**F**) t = 23 h. Continuous red curves are fits to data.

**Figure 3 ijms-23-06007-f003:**
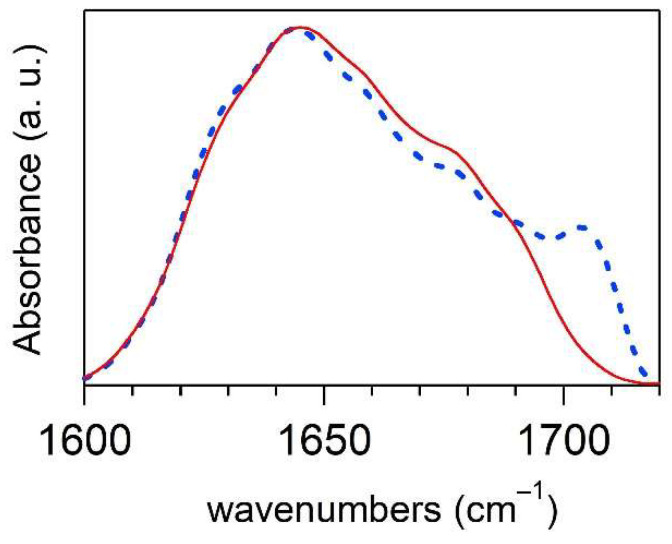
Amide I IR absorbance spectra of Aβ42-cortisol complex at ρ = 1 (c = 400 µM) in 4PBS:1DMSO solution: spectrum of the native state (red curve) and that after 20 h of incubation (blue dashed line).

**Figure 4 ijms-23-06007-f004:**
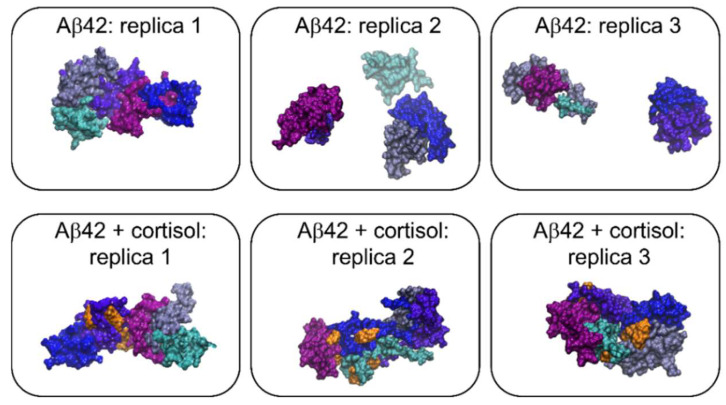
Cartoons of the last configuration (after 200 ns) of each simulated system: in the upper row systems in the absence of cortisol, in the lower row in the presence of cortisol. Blue, cyan, light violet, violet, and purple indicates Aβ42 monomers, whereas cortisol molecule is represented in orange.

**Figure 5 ijms-23-06007-f005:**
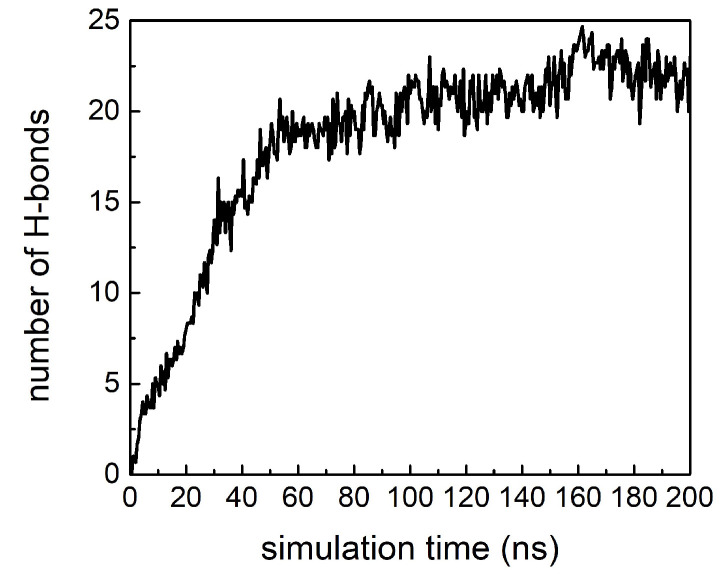
Time evolution of number of H-bonds among Aβ42 and cortisol molecules averaged over the three simulated replicas.

**Figure 6 ijms-23-06007-f006:**
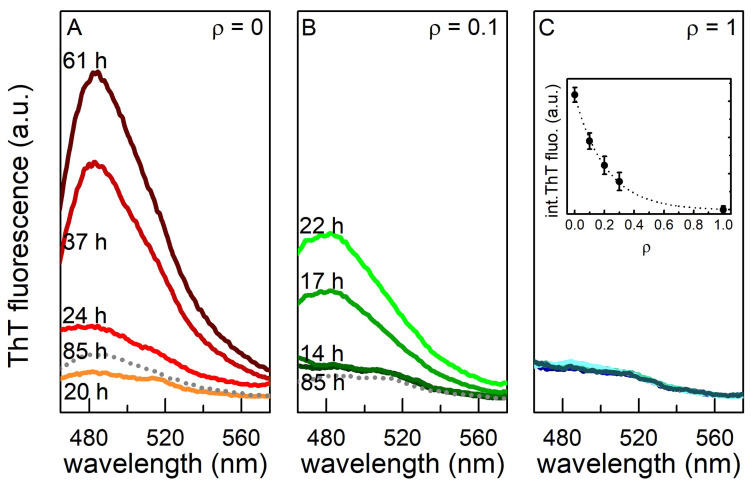
ThT fluorescence spectra during the formation of Aβ42 fibrils at 5 µM concentration in 4PBS:1DMSO buffer at (**A**) ρ = 0, (**B**) ρ = 0.1, and (**C**) ρ = 1. Data are reported on the same intensity scale. In the inset of panel C: maximum integrated ThT fluorescence intensity for each investigated sample as a function of ρ with an exponential fit as a guide for the eye.

**Figure 7 ijms-23-06007-f007:**
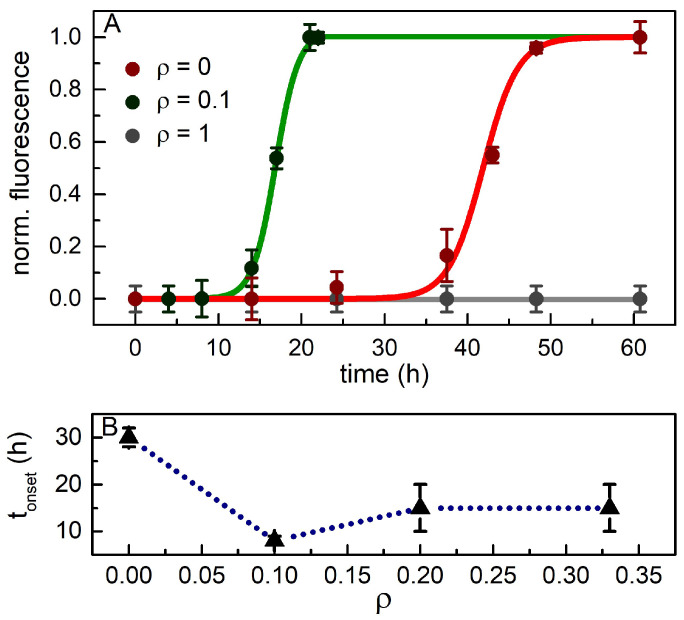
(**A**) Normalized ThT fluorescence intensity as function of incubation time for three selected ρ values with respective Boltzmann fits and errors estimated from repeated measurements; (**B**) concentration dependence of $t_{onset}$: curve is a guide for the eye (data and fits for ρ = 0.2 and ρ = 0.33 are reported in Appendix A). Note that for fluorescence measurements, the Aβ42 concentration is 100 times lower than in the IR experiments; thus, the fibrillation rates scale accordingly.

**Figure 8 ijms-23-06007-f008:**
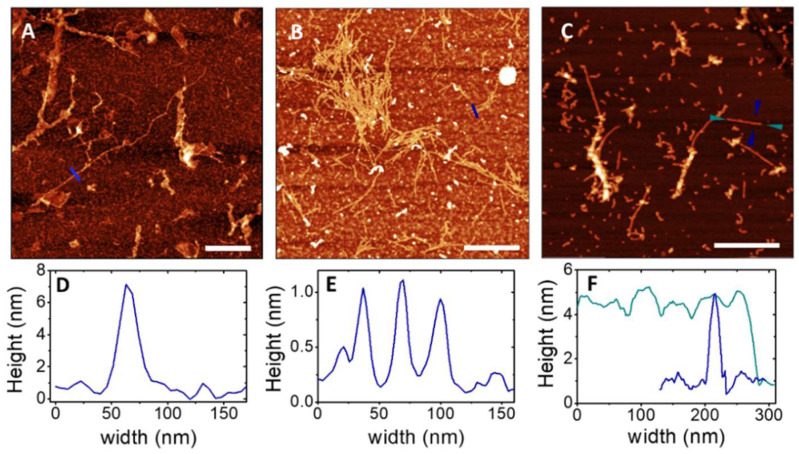
AFM images (height sensor channel) of native Aβ42 sample (ρ = 0) after (**A**) 40 h (**B**,**C**) and 78 h of incubation in 4PBS:1DMSO and (**D**–**F**) corresponding height profiles traced on the marked sections. Bars represent 0.5 µM.

**Figure 9 ijms-23-06007-f009:**
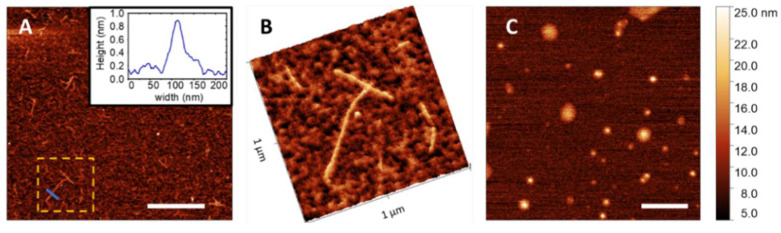
AFM images (height sensor channel) of Aβ42 samples at (**A**) ρ = 0.1 (bar = 1 µM), with (**B**) 3D map and height profile (inset) of the marked region and section, and at (**C**) ρ = 1, bar = 0.5 µM).

**Table 1 ijms-23-06007-t001:** Details of sample preparation at variable cortisol-Aβ42 concentration ratio.

Sample	Aβ42 Volume (Initial Concentration of 500 µM)	Cortisol Volume (Initial Concentration of 500 µM)	4PBS:1DMSO Volume	Aβ42 Final Concentration
ρ = 0	20 µL	0 µL	1980 µL	5 µM
ρ = 0.1	10.9 µL	1.1 µL	1188 µL	4.5 µM
ρ = 0.2	10 µL	2 µL	1188 µL	4.2 µM
ρ = 0.33	9 µL	3 µL	1188 µL	3.8 µM
ρ = 1	6 µL	6 µL	1188 µL	2.5 µM

**Table 2 ijms-23-06007-t002:** Compositions of the six simulated systems.

System Name	Composition
Aβ42: replica 1	5 Aβ42 + 55625 H_2_O + 15 Na^+^
Aβ42: replica 2	5 Aβ42 + 55649 H_2_O + 15 Na^+^
Aβ42: replica 3	5 Aβ42 + 55625 H_2_O + 15 Na^+^
Aβ42 + cortisol: replica 1	5 Aβ42 + 5 cortisol + 55574 H_2_O + 15 Na^+^
Aβ42 + cortisol: replica 2	5 Aβ42 + 5 cortisol + 55567 H_2_O + 15 Na^+^
Aβ42 + cortisol: replica 3	5 Aβ42 + 5 cortisol + 55561 H_2_O + 15 Na^+^

## Data Availability

Not applicable.

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
