# Peer review of "Influence of Cortisol on the Fibril Formation Kinetics of Aβ42 Peptide: A Multi-Technical Approach"

_ijms, 2022, doi:10.3390/ijms23116007_

Round 1

Reviewer 1 Report

The paper deals with the molecular interaction of cortisol and Aβ42 with special attention to the kinetic of amyloid formation. The fibril formation is followed by ThT test while the morphological and conformational changes are characterized by AFM and IR. MD simulation is used to describe the early stage of peptide-cortisol interaction.

The application of various techniques allows getting an overall picture on the processes although the authors have to face to some experimental limitations. The IR investigation could not be performed for peptide-cortisol complex (line 160). This way that study does not contain novel information. Considering the interesting cortisol concentration dependence of fibril formation obtained by ThT measurement it is surprising that the MD simulation was only performed for the case of ρ=1.
The medium for peptide or peptide-cortisol complex is DMSO for IR, water for MD and 20% DMSO/water for ThT. Those differences might or might not influence the results.
The AFM results in the present form do not say more than the absence or presence of fibrils for the cases of high and low concentrations of cortisol. Further analysis of several images recorded for a given system or images taken at various incubation times might shed some light on the development of filament structures.

Small remarks

The title is about the fastening of Aβ42 fibril formation, however the main topic of the report is the role of cortisol in the kinetic of fibril formation emphasising similarly the opposite effect of low and high concentrations.

The effect of high cortisol content is mentioned neither in the Abstract.

Line 265
What time is the end of incubation?

Table S1
The numbers are ratios not %

Fig S5
The measured points of system ρ=0.2 fall reasonably well to the fitted curve of ρ=0.33. The missing points in the time range of 10-25h in Fig S5A resulted in the different sigmoidal curve and – in my opinion - misleading characteristic time values.

Author Response

see file attached

Reviewer 2 Report

Nucara A. et al. describe Ab42 fibril formation kinetics induced by a corticosteroid, cortisol. There are some concerns about this manuscript. Specific points are described below; thus, this reviewer strongly recommends that this manuscript be carefully revised.

Major points:

  1. The authors describe that (i) Cortisol concentration increases during stress conditions and (ii) Cortisol excess is often involved in the early onset of AD. However, it is ambiguous the relationship between cortisol excess and the Abeta amyloid inhibitory mechanism by norepinephrine as refereed in refs. 23 and 24. Furthermore, the blood cortisol concentration is known to be in a range of 5-19 ug/dL that corresponds to 0.14-0.52 uM cortisol; however, such values in addition to the concentration of Abeta peptide in the blood are not fully Why did they use 2.5 uM cortisol as the maximum? The validity of the conditions used is not explained. At least, it should be clearly explained in the introduction.

  1. The IR analysis shown in Fig. 1 was performed in DMSO solvent; however, the PBS and DMSO mixture is used under the cortisol-coexisting condition that is the most important in this manuscript. It is difficult to understand the significance of Fig. 1, especially the relationship with cortisol.

  1. Lines 322-327, the conclusion is inappropriate.

  1. In Figure S1, they performed multi-Gaussian fitting; however, why did they perform it for only 1 condition (Fig. 2A). The analysis for the raw spectrum in 2B and C should also be performed. Moreover, the residual plot between the raw and fitted curves should be represented. How about showing the chi-square values and discussing the validity of each fitting value?

  1. In Fig. 5B, I was very surprised that the errors look entirely the same among all the plotted points. Is it true? Although this journal may not rule out, I believe that all the analytical values used in this figure should be presented as a supplemental figure/table.

  1. In 6 and 7, there is no quantitative and statistical analysis (e.g., reproducibility), just the typical line plot was shown.

Minor points:

The color in Fig. 3 is not explained.

Author Response

see file attached

Round 2

Reviewer 2 Report

Point 1

[My previous comment]

Furthermore, the blood cortisol concentration is known to be in a range of 5-19 ug/dL that corresponds to 0.14-0.52 uM cortisol; however, such values in addition to the concentration of Ab peptide in the blood are not fully explored. Why did they use 2.5 uM cortisol as the maximum? The validity of the conditions used is not explained. At least, it should be clearly explained in the introduction.

[Author’s reply]

The cortisol concentration in the 0.14-0.52 uM range represents the basal level. On the other hand, the AD onset is correlated to stress conditions that are characterized by a steep increase of cortisol beyond the basal levels. It must be added that there is a large variety of opinion in literature for the cortisol basal level (see for instance Widmer, I.E., et al., Cortisol response in relation to the severity of stress and illness. J. Clin. Endocrinol. Metab. 2005, 90, 4579-4586). In the present paper, we investigated the cortisol concentrations that represent either high basal levels (0.45 uM, 0.83 uM) or peak levels (1.25 uM, 2.5 uM) and provide a broad scenario of border or stress conditions. In this regard, we added a few clarifying sentences in the Introduction of the revised version of the manuscript.

[Reviewer’s comment in this round]

Although I understand these authors’ claims, this referee cannot agree that the authors derive experimental conditions based on qualitative concepts without including reported typical values in the introduction in your manuscript.

Point 2

[My previous comment]

In 6 and 7, there is no quantitative and statistical analysis (e.g., reproducibility), just the typical line plot was shown.

[Author’s reply]

We thank the reviewer for raising this point. As discussed in the response to Reviewer 1, we performed a deep statistical analysis of the images reported in Figures 6 and 7 of the original manuscript and on the many several images collected on the samples, to get a quantitative and robust result for the mean size of the typical structures observed in the sample, at the different investigated conditions. We discussed the main results in the revised version of the manuscript, and we added a detailed description in the Supplementary Materials.

[Reviewer’s comment in this round]

As same as my concerns described in point 1, certainly various quantitative values are presented in the supplements. However, I cannot shake off the impression that it is difficult for me to understand, in a direct fashion, that a single reading of the manuscript results could lead to such an interpretation. Likely, this might be because the authors have a different writing policy for manuscripts compared to mine. However, as a paper presented to more general readers, I believe there is room for improvement.

Point 3

[My previous comment]

In Fig. 5B, I was very surprised that the errors look entirely the same among all the plotted points. Is it true? Although this journal may not rule out, I believe that all the analytical values used in this figure should be presented as a supplemental figure/table.

[Author’s reply]

We performed a deeper analysis on these data and amended data in Figure 5. Accordingly,…

[Reviewer’s comment in this round]

I acknowledge that the authors have made improvements, but I cannot understand this as a direct response to my previous comment.

Author Response

see file attached, our replies are provided in bold characters.
